# Smart deworming collar: A novel tool for reducing *Echinococcus* infection in dogs

Shi-Jie Yang[1], Ning Xiao[1,2], Jing-Zhong Li[3], Yu Feng[4], Jun-Ying Ma[5], Gong-Sang Quzhen[3], Qing Yu[1], Ting Zhang[1], Shi-Cheng Yi[6], Zhao-Hui Luo[3], Hua-Sheng Pang[3], Chuang Li[6], Zhuo-Li Shen[7], Ke-Sheng Hou[7], Bin-Bin Zhang[7], Yi-Biao Zhou[8], Hong-Lin Jiang[8], Xiao-Nong Zhou[1,2]*

1 National Institute of Parasitic Diseases, Chinese Center for Disease Control and Prevention; Key Laboratory of Parasite and Vector Biology, Ministry of Health; National Center for International Research on Tropical Diseases; WHO Collaborating Centre for Tropical Diseases, Shanghai, China, 2 One Health Center, School of Global Health, Chinese Center for Tropical Diseases Research, Shanghai Jiao Tong University School of Medicine, Shanghai, China, 3 Tibet Center for Disease Control and Prevention, NHC Key Laboratory of Echinococcosis Prevention and Control, Lhasa, China, 4 Department of Parasitic Diseases, Gansu Center for Disease Control and Prevention, Lanzhou, China, 5 Qinghai Institute for Endemic Disease Prevention and Control, Xining, China, 6 Shanghai Yier Information Technology Co., Ltd, Shanghai, China, 7 Hezuo Center for Disease Control and Prevention, Hezuo city, Gansu province, China, 8 Department of Epidemiology, School of Public Health, Fudan University, Shanghai, China

* zhouxn1@chinacdc.cn

**Data Availability Statement:** National Institute of Parasitic Diseases, Chinese Center for Disease Control and Prevention is the owner of the Data. Any access to the deidentified study dataset would require relevant approvals from National Institute

## Abstract

Echinococcosis is a serious zoonotic parasitic disease transmitted from canines to humans and livestock. Periodic deworming is recommended by the WHO/OIE as a highly effective measure against echinococcosis. However, manual deworming involves significant challenges, particularly in remote areas with scarce resources. The insufficient awareness delivering praziquantel (PZQ) baits for dogs leads to low compliance rate. The aim of this study was therefore to develop a novel smart collar for dogs to address these challenges. We developed a smart Internet of Things (IoT)-based deworming collar which can deliver PZQ baits for dogs automatically, regularly, quantitatively with predominant characteristics of being waterproof, anti-collision, cold-proof and long life battery. Its performance was tested in two remote locations on the Tibetan Plateau. A cross-sectional survey was conducted to evaluate the compliance of the dog owners. Further, a randomized controlled study was performed to evaluate the difference between smart-collar deworming and manual deworming. The collar's effectiveness was further assessed on the basis of Generalized Estimation Equations (GEE). The testing and evaluation was done for 10 smart deworming collars in factory laboratory, 18 collars attached for 18 dogs in Seni district, Tibet Autonomous Region, China, and 523 collars attached for 523 dogs in Hezuo city, Gansu province, China. The anti-collision, waterproof, and coldproof proportion of the smart collars were 100.0%, 99.5%, and 100.0%, respectively. When compared to manual deworming, the dogs' risk of infection with *Echinococcus* on smart-collar deworming is down to 0.182 times (95% CI: 0.049, 0.684) in Seni district and 0.355 (95%CI: 0.178, 0.706) in Hezuo city, the smart collar has a significant protective effect. The owners' overall compliance rate to attach the smart collars for their dogs was 89%. The smart deworming collar could effectively reduce the dogs' risk of infection with *Echinococcus* in dogs, significantly increase the deworming

of Parasitic Diseases, Chinese Center for Disease Control and Prevention and the support of all authors. Point of contact: Ms Gloria Xiong, xiongyh@nipd.chinacdc.cn.

**Funding:** SJY was continuously supported by the Ganzi Prefecture Workgroup Projects for Echinococcosis Prevention and Control (the serial numbers are 2016-07, 2017-06, 2018-02 and 2019-02 respectively), which was initiated by China CDC to advance the echinococcosis control program in 2015 in the Qinghai-Tibet Plateau. JZL received the award from the NHC Key Laboratory of Echinococcosis Prevention and Control Project (Supported by the Non-profit Central Research Institute Fund of Chinese Academy of Medical Sciences, 2019PT320004), and JYM received the Key R & D Transformation Projects (Supported by Science and Technology Committee of Qinghai Province, 2020-SF-133). The funders had no role in study design, data collection and analysis, decision to publish, or preparation of the manuscript.

**Competing interests:** The authors have declared that no competing interests exist.

frequency and coverage and rapidly remove worm biomass in dogs. Thus, it may be a promising alternative to manual deworming, particularly in remote areas on the Tibetan Plateau.

## Author summary

Echinococcosis remains a critical but neglected zoonotic parasitic disease transmitted between canines and livestock or wild rodents. Dogs play a key role in *Echinococcus granulosus* sensu lato (s.l.) and *E. multilocularis* transmission, dual infection also occurs in dogs in co-endemic regions in China. The initial egg-production phase occurs over a span of 34–58 days (*E. granulosus*) or 28–35 days (*E. multilocularis*) following infection. The free-living eggs, voided with faeces of the definitive host, can withstand extreme weather conditions and remain viable for 240 days (*E. granulosus*) or 41 months (*E. multilocularis*) in the environment. Removal or reduction of the worm biomass in dogs will have the greatest and fastest effect in terms of reducing active transmission. In China, although significant efforts have been expended to achieve monthly manual deworming, the actual frequency and coverage thereof remain low; therefore, *Echinococcus* spp. is still highly prevalent among dogs. Moreover, echinococcosis transmission is still rampant, as the heavy disease burden suggests. We propose a novel, smart, Internet of Things (IoT)-based deworming tool that can deliver PZQ baits to dogs regularly and automatically. It could increase the deworming frequency and coverage significantly, reduce the risk of infection by down to 0.182–0.355 times, and prevent canine infections by removing the worm biomass in dogs rapidly. This deworming collar could also potentially prevent the transmission of echinococcosis from dogs to humans and livestock completely. It may be an excellent alternative to existing manual deworming methods, and the difficulties associated with performing deworming in remote areas with scarce resources can be overcome. Since the discovery of PZQ as the most effective antiworm drug, few breakthroughs have been achieved in terms of novel tools and technologies for the control of echinococcosis. Over the last 50 years, echinococcosis control measures have lagged to keep up with the World Health Organization (WHO) roadmap for the elimination of the disease owing to the practical difficulties in remote areas with scarce resources as well as the lack of promotion of new technologies. We expect the proposed smart deworming collar to herald the development of more innovative technologies for controlling echinococcosis by accelerating the elimination of the disease.

## Introduction

Echinococcosis, a severe zoonotic parasitic disease[1], has been listed as a neglected tropical disease by the WHO, and it was ranked among the top three of 24 global foodborne parasitic diseases by the Food and Agriculture Organization of the United Nation (FAO) /WHO in 2010[2,3]. Human echinococcosis presents mainly in two forms: cystic echinococcosis (CE), caused by *Echinococcus granulosus* sensu lato (s.l.) infection, and alveolar echinococcosis (AE), caused by *E. multilocularis* infection. *E. multilocularis* is distributed in the northern hemisphere across 36 countries and regions, while *E. granulosus* is universally distributed with the exception of Antarctica[4,5]. In China, a national epidemiological survey showed that CE is endemic in at least 368 counties of nine provinces (autonomous regions) in northwest China, and it is co-endemic with AE in 115 of these counties[6].

 Previous studies have estimated the global burden of CE to be approximately 1 million disability-adjusted life years (DALYs), of which China accounts for 40%[7]. Global human-

associated direct and indirect costs due to *E. granulosus* infection have been estimated at $764 million, while livestock-associated costs due to *E. granulosus* infection have been estimated at over $2 billion per annum, of which China accounts for a significant share[7]. The latest estimates suggest an annual global incidence of at least 188,000 new CE cases, resulting in 184,000 DALYs, i.e., 0.98 DALYs per case, with 91% of the cases and 95% of the DALYs occurring in China each year [8]. The estimated prevalence of echinococcosis in the nine epidemic provinces in China was 0.51% between 2012 and 2016; the top three provincial prevalence levels for CE and AE combined were 1.26%, 1.65%, and 1.71% in the Qinghai Province, Sichuan Province, and Tibet Autonomous Region, respectively [6]. China remains the most serious endemic region, where the transmission of both CE and AE persists. Moreover, the disease burden in China is predominant among communities with scarce resources, particularly on the Tibetan Plateau.

Existing field epidemiological investigations have proven that dogs are the most definitive hosts for *E. granulosus* [9]. Additionally, the role of dogs in the transmission of *E. multilocularis* appears to be significant on the Tibetan Plateau in northwest China [10,11]. Dual infection involving both *E. granulosus* and *E. multilocularis* also occurs in dogs in co-endemic regions [12]. Many studies have clearly illustrated that periodic deworming of dogs is highly effective for decreasing the prevalence of *Echinococcus* spp. [13–17]. On the contrary, significant challenges remain regarding preventing or mitigating the progress of the disease. Even in wealthy countries such as New Zealand, Spain, and Chile, sustained efforts are required over a period of 10–50 years [13]. Moreover, high capital and compliance rates are necessary to achieve success. In China, from 2006, the PZQ was employed in a monthly deworming programme to control the transmission of canine echinococcosis [18,19]. However, its administration is extremely difficult, particularly in scattered nomadic communities inhabiting the Tibetan Plateau. The high altitudes, harsh climate, unique religious and cultural practices, insufficient access to all owned dogs, low socio-economic status, and poor overall hygiene exacerbate the difficulty in implementing such deworming measures[13,20]. It is also difficult to obtain real data on the deworming coverage and frequency for dogs. A survey in Xinjiang, where the monthly deworming programme was initiated from 2010 onward, shows that 36·8% of the dog owners had never dosed their dogs, and only 22% of the dogs had been dosed prior to testing [21]. Another survey on PZQ deworming of domestic dogs showed that only 30 of the 138 dog owners (21.7%) dewormed their dogs once a month [22]. The deworming frequency and coverage were far lower than those recommended by the WHO/World Organisation for Animal Health (OIE), i.e. '4–8 times per year' and 'at least >90% of registered dogs', respectively [18,23]. Such difficulties are responsible for the relatively high prevalence of *Echinococcus* spp. among dogs in China, i.e. 2.96%, 3.03%, 4.91%, 7.3%, and 13.0% in Sichuan, Ningxia, Gansu, Tibet Autonomous Region, and Qinghai, respectively [6,20,24]. By contrast, the acceptable prevalence of canine echinococcosis recommended by the WHO is <0.01% after an 'attack' phase involving 5–10 years of regular deworming for registered dogs[18]. Therefore, in China, particularly in the remote epidemic areas on the Tibetan Plateau, there exists an urgent need for a novel tool to deliver PZQ baits for dogs to increase the deworming frequency and coverage, remove worm biomass, reduce the egg abundance in the environment, and ultimately prevent the transmission of *Echinococcus* spp. in rural areas from dogs.

## Methods

### Ethics statement

This study was approved by Laboratory Animal Welfare & Ethics Committee (LAWEC), National Institute of Parasitic Diseases of China CDC (Project NO. IPD -2020-26), all dog owners provided informed written consent both in the local languages (Tibetan) and Chinese.

## Design of the deworming device

The initial design of the deworming device was performed from September 2016 to December 2017, and its 3D model (Fig 1A) was constructed. The required modules, including PZQ delivery, data exchange and communication, power control, GMS+GPS, and motor, were designed on the basis of IoT (Fig 1B)[24]. we optimised and upgraded it (Outer diameter:224mm, Inner diameter: 131mm, Thickness: 39mm, Weight: 400g including 12 PZQ baits; Fig 1C) from January 2018 to June 2019. The smart deworming device can deliver PZQ baits for dogs automatically, regularly, quantitatively with predominant characteristics of being waterproof, anticollision, and cold-proof to ensure it runs well in the harsh climate[25]. A series of tests were also conducted for the production process, to assess performance, functionality, and data exchange, using a remote management system (RMS, a platform specially developed for

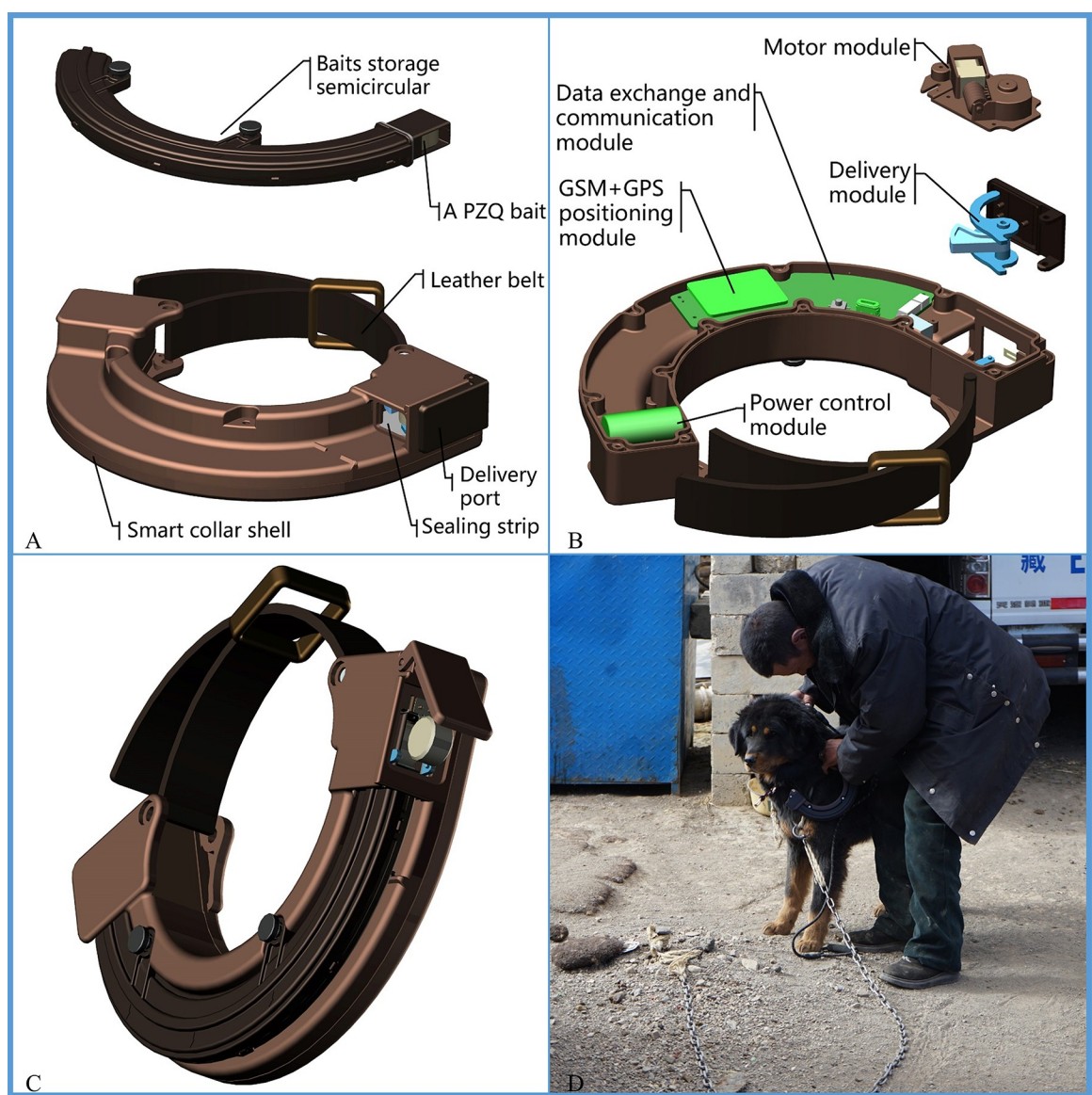

**Fig 1.** Development and field tests of smart deworming collar (A: 3D stacked graph of smart collar; B: Embedded modules for smart collar; C: Overall shape of Smart collar; D: Recovery of collars in July 2019 in Seni district after they had been attached for a year).

managing smart deworming devices), the RMS will automatically identify and analyze the related information and remotely monitor the status of the collar in real time without having to enter the scene[26].

### Pre-field trial

In order to understand the various features of the deworming device, including anti-collision test, waterproof test, accelerated coldproof test and battery endurance test, ten smart deworming collars were produced in the first batch and tested in factory laboratory in Shanghai. We conducted free-fall tests according to the basic environmental testing procedure requirements (IEC 60068-2-32:1990 and GB/T2423.8–1995, 1200mm height, 6 degrees of freedom, free fall, cement floor or floor tile), and followed the IPX6 standards (GB4208-2017 and IEC60529-2013) to carry out waterproof tests, the collars were sprayed from all directions under the tap for a minute. According to the low-temperature test standard for civil equipment (GB/T2423.1 Environmental Test for Electrical and Electronic Products), we placed the collars in a freezer for over 72 h and maintained the temperature below –18˚C for accelerated coldproof and battery endurance tests (see S1 Video). The anti-collision proportion (the ratio of the actual amount of anti-collision collars to the amount that should be collision avoidance), waterproof proportion (the ratio of the actual amount of waterproof collars to the amount that should be waterproof), and coldproof proportion (the ratio of the actual amount of cold-proof collars to the amount that should be protected from the cold) were considered for evaluating the smart collars.

### Field trial

**Field settings.** The field trials to further verify and evaluate the performance and function of the collar were performed in Seni district, Nagqu city, Tibet Autonomous Region of China, and Hezuo city, Gannan Tibet Autonomous Prefecture, Gansu Province of China. Both of them are pastoralist communities with about 144,000 and 95,000 people respectively. From July 2018 to June 2019, the first field trial was conducted to evaluate the effectiveness of the smart deworming collar in Seni district, where the average altitude is >4500m and the average temperature is below -2.2˚C, both CE and AE are co-endemic with an echinococcosis prevalence of 3.68% in humans and 7.14% in dogs. From September 2019 to August 2020, similar field trial was conducted in Hezuo city, where the average altitude is >3000 m and the average temperature is between –0.5˚C and 3.5˚C, with an echinococcosis prevalence of 0.13% in humans, 3.18% in dogs, and 3.43% in livestock [27]. The working temperatures and battery voltages of each smart deworming collar were recorded and uploaded to the RMS in real time. The automatic delivering PZQ proportion represents the ratio of the actual number of PZQ baits automatically delivered to the number of baits that should be delivered, the latter equals the product of the number of collars and number of deliveries (i.e. once a month or 12 times a year, n = 12 in Seni district and n = 523 in Hezuo city) set initially in the RMS. The collar positioning proportion refers to the ratio of the actual positioning numbers to the set total positioning numbers, in Seni district (resp. Hezuo city), the collars were set to upload the locations of the dogs with the smart deworming collars once every seven days. The deliver PZQ reminding proportion represents the ratio of the actual number of reminding to the number of that should be reminded, the latter equals the product of the number of collars and number of reminding (i.e. once a month or 12 times a year, n = 18 in Seni district and n = 523 in Hezuo city); The failure proportion of the collars represents the ratio of the actual number of failure to the number of that should be run well, those faults include the missing PZQ bait delivery, missing report for positioning, and mistakes in deworming reminders; the fault report

proportion represents the ratio of the actual reported number of failure to the number of that should be reported, a high fault report proportion represents good operating conditions of the smart collars and the RMS.

## Grouping and sample size

All registered dogs in Seni district and Hezuo city were included in the study. The exclusion criterias include: (1) pregnant dogs; (2) weight is less than 5kg or age is less than 1 months; (3) size is too big to be attached. The dogs in Seni district and Hezuo city are divided into smart deworming group and manual deworming group respectively. We estimated the total sample size by the Walters normal approximation method according to the two samples (two groups) parallel control design, $P_m \approx 7.14\%$ (positive rate of *Echinococcus* antigen in manual deworming group, its value comes from the literature), $P_s \approx 0.01\%$ (positive rate of *Echinococcus* antigen we expect to reach according to the literature [18]) in Seni district and $P_m \approx 3.43\%$ (manual deworming group), $P_s \approx 0.01\%$ (smart deworming group) in Hezuo city, $\alpha = 0.05, 1-\beta = 0.8$, $N_s$:$N_m \approx 3:1$, the estimated sample should be greater than 608 (456:152), and the total sample size was thus determined to be 741 (541:200).

In Seni district, 18 smart deworming collars were used for field testing of performance and function, we numbered 6,882 registered dogs one by one and used SPSS version 20 (IBM Corp, Armonk, NY, USA) to generate a group each of 18 dogs (distributed in 14 villages of 12 townships) as the smart deworming group and the manual deworming group via accurate random sampling. In Hezuo city, a total of 523 smart deworming collars were tested and using the same software, we generated a group of 523 dogs (distributed in 48 villages of 12 townships) as the smart deworming group and the manual deworming group (182 dogs, distributed in 43 villages of 12 townships) via accurate random sampling from the 7,463 registered dogs.

**Interventions.**  According to the generated sampling lists of dogs in Seni district and Hezuo city, the staff attached the smart deworming collars to 541 dogs (523 in Hezuo city and 18 in Seni district) in the smart deworming group, matched the information of the dogs and owners with the collars, and uploaded the data to the RMS. The steps of attaching smart deworming collars for dogs is simple and easy to operate[25]. The RMS provides various options of deworming frequencies, deworming periods, and deworming doses for each dog [26]. The PZQ baits (100 mg of PZQ per bait, Batch: 20180613), manufactured by Beijing Zhongnong Warwick Pharmaceutical Co., Ltd., Beijing, China, were prepared into microcapsules by embedding PZQ, the favourite foods (meat, fatty oil, etc.) for carnivores were selected as the core to develop an attractive core suitable for greatly concealing the bitter taste and unpleasant odor of PZQ[28]. The dosage of the PZQ baits for each dog in attached smart deworming collar is set in accordance with the instruction of the PZQ baits. In order to ensure that the PZQ baits delivered can catch dog's attention to eat, the smart deworming collar was set the reminder function, that is, it will sound a reminder within 3 minutes before the PZQ baits are delivered. The deworming time was set to avoid the smart collar delivering the baits when the dog was running (or walking), in our study, it was set to 6 a.m. on the deworming date mainly also because at this time, the dog is still sleeping and hungry as it has not been fed overnight and is hence more likely to eat the PZQ baits. Moreover, the PZQ bait itself have a tempting flavor to attract dogs eating. The staff observed the first deworming process of a smart collar to confirm that it worked normally, record the information dogs swallowed PZQ bait (30 minutes) and subsequently proceeded to attach the other collars (see the detailed information in S1 Video). Deworming information including the deworming reminder, deworming time and times will be automatically recorded and uploaded onto the RMS[26].

To ensure that the collected data of ingested the PZQ baits are true and effective, we signed a letter of commitment with every owners in order to enhance their compliance, urged the village dewormers (the village administrators) to make telephone calls to owners) at about 5:50am in the morning of the deworming day to remind the owners to observe and record whether their dogs swallowed the bait or not, and the dog owners also were asked to fill in Record Form about dog swallowing in time, take real-time photos, and send the photos to the village administrator within 30 minutes. The collar was removed from dog after 12 months attaching (Fig 1D) and its performance and function were evaluated again. In the manual deworming groups, the corresponding related information of dogs and owners also collected and uploaded to the RMS i.e. 18 dogs in Seni district and 182 dogs in Hezuo city, and the control dogs were manually dewormed once a month. This process typically included delivery of the PZQ baits by local veterinarians to the dog owners, telling them to feed to their dogs once a month on the deworming date. In addition, all dog owners from both groups were given routine health education on the prevention and control of echinococcosis, involving measures such as washing hands frequently and not feeding their dogs with livestock viscera or playing with the dogs.

**Table 1. Deworming times, collected samples and positive samples of dog faeces for two groups in field areas.**

| Field areas | Deworming times | Smart deworming group | | | | Manual deworming group | | | |
|---|---|---|---|---|---|---|---|---|---|
| | | Numbers to be collected | Missing numbers | Actual numbers collected | Number of positive samples (%) | Numbers to be collected | Missing numbers | Actual numbers collected | Number of positive samples (%) |
| Seni district | 0 (before deworming) | 18 | 0 | 18 | 2(11.1) | 18 | 0 | 18 | 2(11.1) |
| | 1st | 18 | 0 | 18 | 3(16.7) | 18 | 0 | 18 | 2(11.1) |
| | 2nd | 18 | 0 | 18 | 0 | 18 | 1 | 17 | 1(5.9) |
| | 3rd | 18 | 1 | 17 | 0 | 18 | 0 | 18 | 2(11.1) |
| | 4th | 18 | 0 | 18 | 0 | 18 | 1 | 17 | 2(11.8) |
| | 5th | 18 | 1 | 17 | 0 | 18 | 1 | 17 | 3(17.6) |
| | 6th | 18 | 2 | 16 | 1(6.3) | 18 | 2 | 16 | 4(25.0) |
| | 7th | 18 | 3 | 15 | 0 | 18 | 3 | 15 | 3(20.0) |
| | 8th | 18 | 0 | 18 | 0 | 18 | 1 | 17 | 2(11.8) |
| | 9th | 18 | 1 | 17 | 0 | 18 | 2 | 16 | 3(18.8) |
| | 10th | 18 | 2 | 16 | 0 | 18 | 2 | 16 | 2(12.5) |
| | 11th | 18 | 2 | 16 | 0 | 18 | 3 | 15 | 1(6.7) |
| | 12th | 18 | 2 | 16 | 0 | 18 | 3 | 15 | 1(6.7) |
| s | 0 (Before deworming) | 523 | 49 | 474 | 16(3.4) | 182 | 5 | 177 | 6(3.4) |
| | 1st | 523 | 37 | 486 | 19(3.9) | 182 | 12 | 170 | 6 (3.5) |
| | 2nd | 523 | 48 | 475 | 1 (0.2) | 182 | 13 | 169 | 6(3.6) |
| | 3rd | 523 | 36 | 487 | 2(0.4) | 182 | 19 | 163 | 7(4.3) |
| | 4th | 523 | 56 | 467 | 1(0.2) | 182 | 12 | 170 | 7(4.1) |
| | 5th | 523 | 81 | 442 | 0 | 182 | 21 | 161 | 5(3.1) |
| | 6th | 523 | 32 | 491 | 0 | 182 | 13 | 169 | 5(3.0) |
| | 7th | 523 | 29 | 494 | 0 | 182 | 11 | 171 | 6(3.5) |
| | 8th | 523 | 42 | 481 | 0 | 182 | 9 | 173 | 5(2.9) |
| | 9th | 523 | 51 | 472 | 1(0.2) | 182 | 12 | 170 | 4(2.4) |
| | 10th | 523 | 38 | 485 | 0 | 182 | 16 | 166 | 5(3.0) |
| | 11th | 523 | 36 | 487 | 0 | 182 | 9 | 174 | 5(2.9) |
| | 12th | 523 | 59 | 464 | 0 | 182 | 8 | 169 | 5(3.0) |

**Faecal sample tests.** To establish a baseline before the commencement of our study, local veterinarians collected the first batch of faecal samples from a total of 741 targeted dogs, as per a printed sampling list. After the smart delivery of the PZQ baits to each dog in the smart deworming group, the canine faecal samples were collected once a month from both groups under different considerations (see Table 1). Each sample was labelled with the assigned number and location and date of sampling and matched with the information of the dog. The data was then uploaded to the RMS. The samples were taken to the authorized laboratory, frozen at −80˚C for at least seven days, and then maintained at −20˚C until examination[29].

The tests were conducted once a quarter. The coproantigen ELISA Kit for Canine was produced by Shenzhen Combined Biotech Co., Ltd., Shenzhen, China. To ensure quality, the testing staff were trained using the same batch of testing kits (Batch: 20180801, the sensitivity is 95.45% and the specificity is 96.97%). The operation steps were followed as per the product manual.

**Questionnaires.** The subject of the questionnaire survey is the owner of the dog wearing the smart collar. Their dogs were randomly selected as wearing a smart collar (in Seni district, 18 people, in Hezuo city, 523 people). If the owner chose was unwilling to attach the smart collar to the dog, the subject of the survey was randomly again selected as the process of selecting dogs wearing smart collar. The questionnaire, both in Chinese and Tibetan, was conducted twice on the dog owners in the beginning and end of study respectively. It included general information of the owner, dog, village, PZQ-administration, continuous attachment with smart collars, the dog's reaction following the collar attachment, and several questions on the owners' attitude and compliance regarding smart collar attachment for the dogs—particularly the reasons for unwillingness to attach or remove the collar. All questionnaires were identified via unique numerical identification, and the data were uploaded on the RMS. To ensure cultural appropriateness of the questionnaire and to guarantee that each question was fully understood, we designed the questionnaire by group of the authors who have language skills in Chinese and Tibetan language, and pre-tested in a small pilot study. The questionnaire was undertaken with each dog owner in intervention group by one of the co-authors who was a native speaker of both Chinese and Tibetan language. we employed the PZQ-swallowing proportion (the ratio of the actual amount of PZQ baits swallowed to the amount that should be swallowed) to evaluate the swallowing of baits by dogs, the consecutive attachment proportion (the ratio of the actual amount of smart collars attached for the dogs to the amount that should be attached) to assess the consecutive attachment, and the dogs' reaction proportion (the ratio of the actual dog amount of reaction on attached smart collars to the dog amount that should respond to smart collars) to evaluate the impact of collars on dogs.

## Data analysis

The database was created and managed using Microsoft Excel version 2016 (Microsoft Corporation, Redmond, WA, USA). SPSS version 20 was used for data analysis. Continuous variables were tested for normal distribution and described as mean and standard deviation (SD), while categorical variables were exhibited as frequency and percentage. For analysis purpose, the result of Echinococcus antigen (positive or negative) in canine faeces at different time points was set as the response variable, and the deworming method (i.e. whether adopting smart collar) as the explanatory variable. GEE was then performed to analyze the deworming effect of smart collar by fitting a logistic model to the repeatedly measured categorical responses to compare the positive rate of Echinococcus antigen between the smart collar group and the control group. Each variable with assignment is shown in Table 2. Differences were tested with two-tailed tests and P<0.05 was considered statistically significant.

**Table 2. Variables and the assignments in the analysis.**

| Variable | Meaning | Assignment | Variable type |
|---|---|---|---|
| ID | The ID number of each subjects | -- | Subject variable |
| Outcome | Results of *Echinococcus* antigen in canine faeces | 1 positive<br>0 negative | Response variable |
| Method | Intervention methods used for deworming | 1 smart collar<br>0 manual deworming | Explanatory variable |
| Time | Time points at which the *Echinococcus* antigen is tested | -- | Within-subject variable |

### Role of the funding source

The funders had no role in study design, data analysis, data interpretation, or writing of the manuscript. All authors had access to all the data in the study and had final responsibility for the decision to submit for publication.

## Results

### Pre-field trial

Ten smart deworming collars were tested directly after being stricken, sprayed, and frozen in Shanghai (see S1 Video). 18 and 523 smart deworming collars were retrieved and detected after 12 months of wearing in dogs in Seni district and Hezuo city respectively. The results have shown that it was fully capable of anti-collision, waterproof, and coldproof performance not only during the experiments in factory laboratory, but also under the harsh climate at remote locations on the Tibetan Plateau, even being attached in dogs for 12 months. The anti-collision, waterproof, and coldproof proportion of 551 smart deworming collars were 100.0%, 99.5%, and 100.0%, respectively (see Table 3).

### Field trial

**Battery endurance and working temperature of smart collar.** The average voltage of the collars fell to 3.95±0.08 V from 4.12±0.03 V (n = 10) after storage for 72 h in a freezer below –18˚C (accelerated test in factory laboratory). In field trials lasting 12 months, the gentle discharge curve ranges are much above the termination voltage, and the collar working temperature is much higher than the lowest ambient temperature attained on the same day (Fig 2A: in Seni district, n = 18; Fig 2B, in Hezuo city, n = 523).

**Automatic PZQ delivery proportion and failure proportion.** As illustrated in Table 4, the evaluation of the smart deworming collar function in the field areas yielded satisfactory results, i.e. a very high bait delivery proportion of 87.8%, low overall failure proportion of 10.0%, and high fault report proportion of 89.8%.

**Compliance rate and continuity proportion of attaching smart collar to dog.** In Seni district, 19 dog owners were instructed to attach the smart deworming collars to their dogs; however, one refused, i.e. the compliance rate (collar attachment willingness rate) was 94.7%

**Table 3. Results of detection on performance of smart deworming collars.**

| Verification time | Detection areas | Total number of collars | Water-proof proportion (%) | Anti-collision proportion (%) | Cold-proof proportion (%) |
|---|---|---|---|---|---|
| 2018.6 | Shanghai | 10 | 9/10(90.0) | 10/10(100.0) | 10/10(100.0) |
| 2018.7–2019.6 | Seni district | 18 | 18/18 (100.0) | 18/18 (100.0) | 18/18 (100.0) |
| 2019.9–2020.8 | Hezuo city | 523 | 521/523 (99.6) | 523/523 (100.0) | 523/523 (100.0) |
| Total | | 551 | 548/551 (99.5) | 551/551 (100.0) | 551/551 (100.0) |

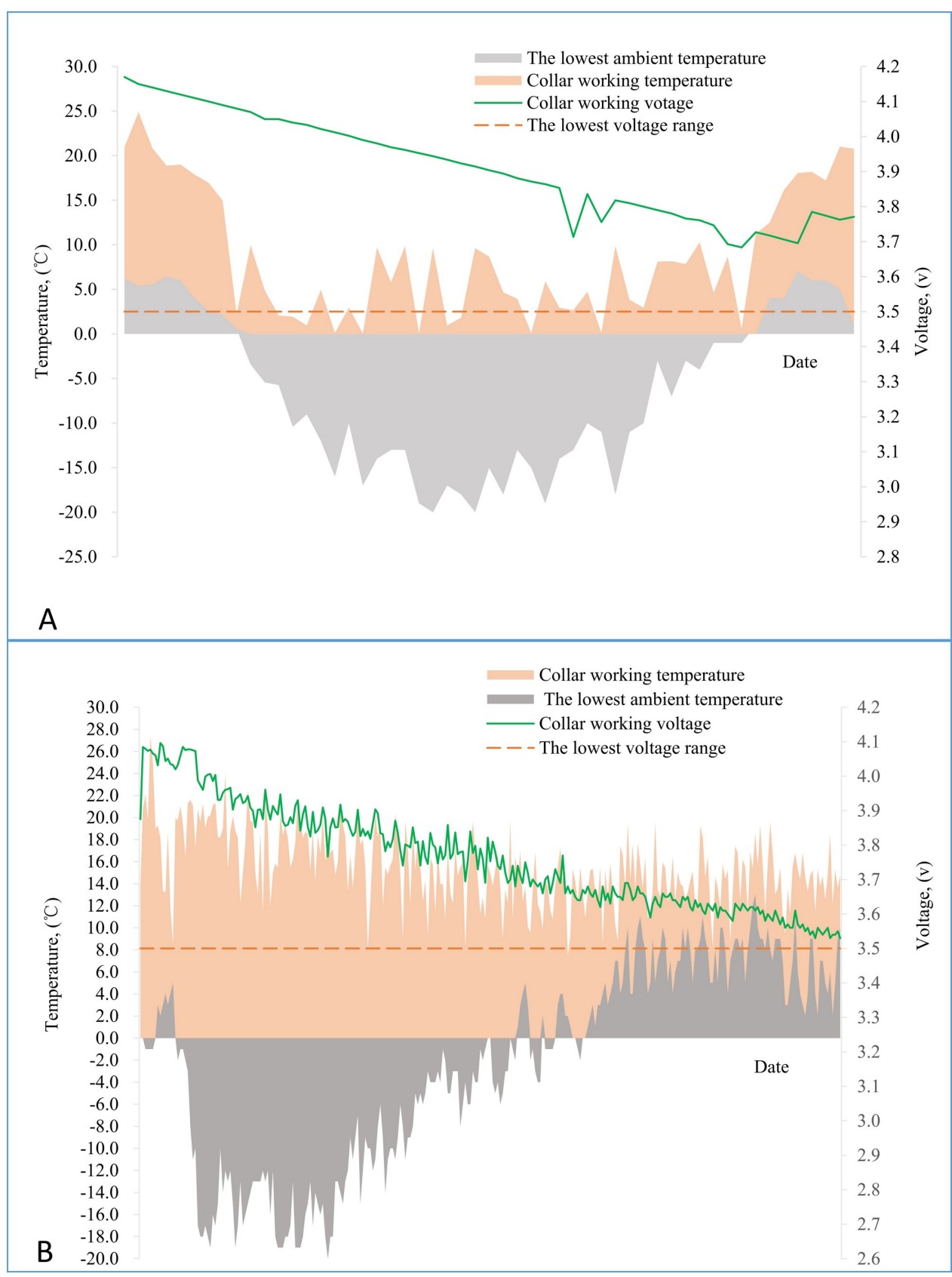

**Fig 2.** Voltage and temperature change of smart collars with ambient temperature (A: from Jul 31, 2018 to Jul 24, 2019, in Seni district, n = 18; B: from sept 24, 2019 to sept 8,2020, in Hezuo city, n = 523).

(18/19), while it was 88.8% (523/589) in Hezuo city. A total of 67 dog owners were reluctant to use the collar, their main concerns as follows: (1) the dogs' incompatibility with the collar (25.4% (17/67)), (2) the impact of the collar on the dog's duties, i.e. protecting property and livestock (47.8% (32/67)), (3) side effects of the PZQ baits (19.4% (13/67)), (4) tightness of the collar (6.0% (4/67)), and (5) the collars effects on the dogs' reproduction (1.5% (1/67)). The collar attachment proportion for six and twelve consecutive months were 92.8% and 85.8%, respectively (see Table 5); a total of 77 smart collars were removed by the owners at the end of 12 months, the reasons mainly included: (1) removal due to irritation and discomfort, 37 (48.1%), (2) removal by the owner during the transition between the pastures (winter) and the ranch (summer) and failure to attach it again, 34 (44.1%), and (3) removal by the dogs themselves, 6 (7.8%). In the manual deworming group, when the owners were asked as to whether their dogs had been dewormed, 81% (162/200) provided an affirmative response. However, when they were asked additional questions regarding the deworming, such as 'where are the PZQ baits' and 'where are the deworming records', the majority (89%, 178/200) displayed awkward expressions; only 29% (47/162) presented the baits, and only 22.2% (32/162) had deworming cards.

**Dogs' reaction to smart collar attachment and deworming reminders.** All 541 dogs exhibited reactions to the collar attachment; in particular, 402 dogs (74.3%) moved in circles, trying to bite or remove the collar. Further, all dogs responded immediately to the PZQ delivery reminder sound (e.g. they stood up, pricked the ears, searched for the source of the sound, and were more alert) (see Table 5).

**Positive rates of Echinococcus antigen in canine faeces.** Table 1 depicts that in smart deworming group in Seni district, from the second deworming, the positive sample of Echinococcus antigen was always maintained at 0 except for the sixth deworming procedure, in Hezuo city, from the second to the fourth deworming procedure, the positive rate of Echinococcus antigen decreased to 0.2%, 0.4%, and 0.2%, in particular, from the fifth deworming, a positive rate of 0% was maintained except for the ninth deworming procedure. However, in manual deworming group, whether it is before or during deworming, in Seni district, the positive samples of Echinococcus antigen were always maintained in the same level, and in Hezuo city, the positive rate of Echinococcus antigen were not significant change.

**Protective effect of the smart collars.** Results based on GEE in Seni district and Hezuo city are shown in Table 6, which presents a significantly lower positive rate in the smart deworming group than that in the manual deworming group, indicating a better deworming effect of smart collar intervention. In Seni district, the risk of Echinococcus antigen tested positive of smart deworming group is 0.182 times that of manual deworming group, with a 95% CI range from 0.049 to 0.684 (P = 0.012), while the risk of positive antigen result is 0.335 times (95% CI: 0.178, 0.706; P = 0.003) when using smart deworming compared to manual

**Table 4. Results of evaluation on function of smart deworming collars in field areas.**

| Evaluation time | Field areas | Automatic delivering PZQ proportion (%) | Collar positioning Proportion (%) | Deliver PZQ reminding proportion (%) | Failure proportion (%) | Fault report Proportion (%) |
|---|---|---|---|---|---|---|
| 2018.7–2019.6 | Seni district | 203/216 (94.0) | 2066/2190 (94.3) | 211/216 (97.7) | 142/2622 (5.4) | 132/142 (93.0) |
| 2019.9–2020.8 | Hezuo city | 5496/6276 (87.6) | 24438/27270 (89.6) | 5774/6276 (92.0) | 4114/39822 (10.3) | 3691/4114 (89.7) |
| Total | | 5699/6492 (87.8) | 26504/29460 (90.0) | 5985/6492 (92.2) | 4256/42444 (10.0) | 3823/4256 (89.8) |

**Table 5. Result of evaluation on smart deworming collar attachment.**

| Tests time | Field areas | PZQ-swallowing proportion (%) | | Consecutive attachment proportion (%) | | Dogs' reaction proportion (%) | |
|---|---|---|---|---|---|---|---|
| | | **In 10 min** | **In 30 min** | **In six months** | **In 12 months** | **Smart collar attachment** | **Deworming reminder** |
| 2018.7–2019.6 | Seni district | 129/203 (63.6) | 154/203 (75.9) | 17/18 (94.4) | 17/18 (94.4) | 16/18 (88.9) | 18/18 (100.0) |
| 2019.9–2020.8 | Hezuo city | 3188/5496 (58.0) | 3902/5496 (71.0) | 485/523 (92.7) | 447/523 (85.5) | 386/523 (73.8) | 523/523 (100.0) |
| Total | | 3317/5699 (58.2) | 4056/5699 (71.2) | 502/541 (92.8) | 464/541 (85.8) | 402/541 (74.3) | 541/541 (100.0) |

deworming in Hezuo city. It means that the smart deworming collar has more significant protective effect than the existing manual deworming.

## Discussion

The life cycle of *Echinococcus* spp. includes (i) the adult stage occurring within the definitive hosts, (ii) protoscolex-producing stage occurring within intermediate hosts, and (iii) free-living eggs in the environment. The average adult life span of *E. granulosus* is approximately 10 months, whereas that of *E. multilocularis* is approximately 3–5 months [29]. High rate of egg production is a continuous process over a 35–90-day period after inoculation in the intestines of the dogs in the cases of both *E. granulosus* and *E. multilocularis* [30,31]. When the developed eggs are voided, they contaminate the environment and are even scattered over long distances by the wind or transmitted through the fur, hoofs, or paws of animals. The maximum survival duration (i.e., maintenance of viability) of *E. multilocularis* eggs is 240 days under autumn/winter conditions [32], whereas that of *E. granulosus* eggs is up to 41 months. The life cycle and egg excretion dynamics of *Echinococcus* spp. show irrefutable evidence regarding the rationality and urgency of initiating a monthly deworming programme for controlling canine echinococcosis, particularly in remote co-endemic regions on the Tibetan Plateau in China, where dual infection of dogs can occur.

Availability of sufficient power supply is a prerequisite for ensuring normal delivery of the PZQ baits as per our proposed smart-collar system. Embedded high-capacity rechargeable lithium batteries and low-power technology were employed to ensure that the collar was always powered and in the most energy-saving state (dormant state) until it was activated for positioning or PZQ delivery by the RMS[28]. The tests show that the batteries discharged their energy stably at low temperatures, while the field verification showed that the discharge curve is relatively smooth and far above the lowest voltage range(the termination voltage, see Fig 2A and 2B). Once the smart collar (charged once a year) were attached to dog, the electric energy is enough to keep the smart deworming collar work well for one year (12 times for delivery PZQ baits), which saves a lot of manpower and makes it possible to transform the manual deworming mode to the smart deworming mode.

**Table 6. The effect of smart collar intervention on deworming.**

| Field areas | Groups | N | $\chi^2$ | P | OR | 95% CI |
|---|---|---|---|---|---|---|
| Seni district | Manual deworming group | 18 | | | Reference | |
| | Smart deworming group | 18 | 6.37 | 0.012 | 0.182 | 0.049, 0.684 |
| Hezuo city | Manual deworming group | 182 | | | Reference | |
| | Smart deworming group | 532 | 8.695 | 0.003 | 0.355 | 0.178, 0.706 |

Note: Deworming 12 times in two groups respectively.

The robustness and sealing of the smart collars against moisture or water are the two key factors limiting performance and functioning; the collars must endure long-term and harsh environments. Fortunately, the results of the anti-collision and waterproof tests as well as the on-site verification provided sufficient evidence that the smart collar did not suffer significant damage and displayed excellent moisture and water resistance in harsh environments.

The RMS records indicate that the working temperature of the collar was much higher than the minimum temperature of the external environment. The possible reasons for this difference are as follows: (1) the dog's body heat transmitted to the collar, (2) the collar itself emitted heat, (3) the dog's fur reduced heat emission, and (4) the dogs stayed in the owner's house or tent with stoves in winter. Such factors allowed for the collars to perform all their basic pre-set functions, such as timed PZQ delivery, positioning, sounding of deworming reminders, and fault reporting in harsh climatic conditions (see Table 4).

The cross-sectional survey showed that the owners' overall compliance rate was 89%, which can be regarded as an indicator of future coverage of smart collars to judge their applicability. Although 39 and 77 collars were removed after being attached for six and twelve months, 76.3% (464/608) ~ 82.6% (502/608) of actual coverage is still a satisfactory value and is close to the coverage of 90% recommended by the WHO/OIE[18,23]. A PZQ delivery proportion of 87.8% and continuous collar attachment proportion of 85.8%~92.8% were recorded by the RMS (see Tables 4 and 5). In the manual deworming group, the manual deworming frequency and coverage were relatively low or unavailable.

A randomized controlled study was conducted to evaluate the difference between smart-collar deworming group and manual deworming group in terms of their protective effectiveness. The baseline positive samples of *Echinococcus* antigen in canine faeces in both the groups was 11.1% in Seni district, and was 3.4% in Hezuo city, respectively (Table 1). After the first deworming procedure in the two smart deworming groups, the positive rate of dog faeces increased from 3.4% to 3.9% in Hezuo city ($\chi^2 = 0.005$, p = 0.943), and the positive samples of *Echinococcus* antigen in canine faeces from 2 to 3 in Seni district. However, this was not significant. Nevertheless, this increase can be regarded as an indicator of potential infection in the dog population that is revealed by deworming with the smart collars. Table 1 shows that the overall positive rate (or samples) of *Echinococcus* antigen in smart deworming group have a decreasing trend with increasing deworming frequency.

Two early studies indicated that higher levels of reinfection following deworming are observed in spring and early winter, and the pressure for infection with *E. granulosus* in dogs varies seasonally owing to the fluctuating frequency of livestock slaughter [11,33]. A similar scenario of reinfection has been investigated in our study. A sudden increase in the positive samples of *Echinococcus* antigen in dog faeces from 0, 0, 0, 0 to 1 and from 1, 2, 2, 3 to 4 after the sixth deworming procedure in the smart deworming group and the manual deworming group, respectively, was observed in Seni district, for sample collection in January 2019. It is noteworthy that the livestock were mainly slaughtered from the end of November to the beginning of December 2018. Considering the average initial onset of egg production [26], we strongly suspect that the reinfection in the dogs occurred because the livestock viscera were fed to the dogs during the slaughter period. In the smart deworming group, owing to the regular and automatic deworming by the smart collars, deworming frequency was guaranteed, and the positive samples of *Echinococcus* antigen decreased rapidly to 0 again. By contrast, in the manual deworming group, the deworming frequency and effects were low; the *Echinococcus* infection in these dogs could not be destroyed in time, resulting in continuous reinfection, increased egg emission, and a high transmission risk in the environment (see Table 1). In Hezuo city, 523 smart collars were attached in September, 2019, and the slaughter period was mainly from the middle to the end of November. Some fecal samples from second, third, and

fourth collection were collected after 30–50 days during the slaughter period; thus, the infection or reinfection rate of the dogs were maintained at a certain level (see Table 1). From the fifth deworming procedure onward, the positive rate of *Echinococcus* antigen in dog faeces decreased to 0%. However, in the manual deworming group, the positive rate of *Echinococcus* antigen increased from the second deworming procedure onward and continued to remain at a high level (see Table 1).

Echinococcosis is a progressively parasitic disease, its transmission is complicated and involves multiple hosts as well as many risk factors. The acceptable prevalence of canine echinococcosis is <0.01% after an 'attack' phase of 5–10 years of regular deworming for registered dogs[18]. The limitations of this study is its relatively short evaluation period, and the actual effect of implementation still needs to be evaluated for a longer time. The weight and size of the smart deworming collar limit its application to all dogs, further optimization and upgrading is urgent to reduce weight and size and make dogs more comfortable. Health education for dog owners should be strengthened to improve their willingness. In addition, the associated economics and potential impact on dogs all need further to be evaluated.

In this study, we employed GEE to further evaluate the protective effect of the smart deworming collars on dogs. The results show that in contrast to manual deworming, smart-collar deworming can reduce the dogs' risk of *Echinococcus* infection by down to 0.182~0.355 times. In conclusion, it is essential to support the application of the smart deworming collar in control programs for echinococcosis.

## Supporting information

**S1 Video. The characteristics for the Smart Deworming Collars.mp4.**
(MP4)

## Acknowledgments

We are grateful to Hezuo city Center for Disease Control and Prevention, Gansu Province, and Seni district Center for Disease Control and Prevention in Tibet Autonomous Region, for providing us the field site, and cooperating with us to complete the field work and questionnaire survey, laboratory testing, and providing basic monitoring data and baseline data to RMS. We also specifically thanks to Dr Jun-Fang Xu for her statistical advices in data handling.

## Author Contributions

**Conceptualization:** Shi-Jie Yang, Ning Xiao, Xiao-Nong Zhou.

**Data curation:** Shi-Jie Yang, Yi-Biao Zhou, Hong-Lin Jiang.

**Formal analysis:** Shi-Jie Yang, Xiao-Nong Zhou.

**Funding acquisition:** Shi-Jie Yang, Jing-Zhong Li, Jun-Ying Ma.

**Investigation:** Shi-Jie Yang, Yu Feng, Gong-Sang Quzhen, Qing Yu, Zhao-Hui Luo, Hua-Sheng Pang, Zhuo-Li Shen, Ke-Sheng Hou, Bin-Bin Zhang.

**Methodology:** Shi-Jie Yang, Ning Xiao, Yu Feng, Jun-Ying Ma, Gong-Sang Quzhen, Ting Zhang.

**Project administration:** Shi-Jie Yang, Ning Xiao, Jing-Zhong Li, Xiao-Nong Zhou.

**Software:** Shi-Jie Yang, Shi-Cheng Yi, Chuang Li, Yi-Biao Zhou, Hong-Lin Jiang.

**Supervision:** Shi-Jie Yang, Ning Xiao, Xiao-Nong Zhou.

**Validation:** Shi-Jie Yang, Ning Xiao.

**Visualization:** Shi-Jie Yang, Shi-Cheng Yi.

**Writing – original draft:** Shi-Jie Yang.

**Writing – review & editing:** Shi-Jie Yang, Ning Xiao, Jing-Zhong Li, Yu Feng, Jun-Ying Ma, Gong-Sang Quzhen, Qing Yu, Ting Zhang, Shi-Cheng Yi, Zhao-Hui Luo, Hua-Sheng Pang, Chuang Li, Zhuo-Li Shen, Ke-Sheng Hou, Bin-Bin Zhang, Yi-Biao Zhou, Hong-Lin Jiang, Xiao-Nong Zhou.

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
