## [Decision Letter · Decision Letter 0]

23 Dec 2020

Dear Zhou,

Thank you very much for submitting your manuscript "Smart Deworming Collar: A Novel Tool for Reducing Echinococcus Infection in Dogs" for consideration at PLOS Neglected Tropical Diseases. As with all papers reviewed by the journal, your manuscript was reviewed by members of the editorial board and by several independent reviewers. In light of the reviews (below this email), we would like to invite the resubmission of a significantly-revised version that takes into account the reviewers' comments. 

We cannot make any decision about publication until we have seen the revised manuscript and your response to the reviewers' comments. Your revised manuscript is also likely to be sent to reviewers for further evaluation.

Sincerely,

Paul Robert Torgerson

Associate Editor

Adriano Casulli

Deputy Editor

Reviewer's Responses to Questions

**Key Review Criteria Required for Acceptance?**

**Methods**

-Are the objectives of the study clearly articulated with a clear testable hypothesis stated?

-Is the study design appropriate to address the stated objectives?

-Is the population clearly described and appropriate for the hypothesis being tested?

-Is the sample size sufficient to ensure adequate power to address the hypothesis being tested?

-Were correct statistical analysis used to support conclusions?

-Are there concerns about ethical or regulatory requirements being met?

Reviewer #1: Prior to describing the appearance of the device, it would be helpful to briefly describe the purpose of the collar and how it is used. Are the praziquantel baits flavored? Are dogs supposed to notice that a bait has fallen out of the collar and eat it without human assistance? How can one confirm that the bait was actually eaten by the dog wearing the collar, especially if the dog is moving when the bait drops and/or a human isn’t present?

Are both Seni and Hezuo pastoralist communities? What is the approximate size of these communities?

Please briefly describe the registration methods/requirements for dogs in the study area. Were there any inclusion/exclusion criteria for enrollment in this study (e.g., size/weight of the dog)? 

Briefly provide the instructions given to the owners of dogs in the collar and non-collar groups regarding deworming, record keeping, etc. It was not clear if owners in the manual deworming group were given instructions to deworm the dog or if a local veterinarian was responsible for dog deworming and, if both occurred, how the analysis took this into account.

Please elaborate on what dog and owner information was initially collected and uploaded to the RMS. Please also clearly provide a description of what data the collar collects and how this information is downloaded from the collar (and the frequency at which it is downloaded). 

Is there a published reference for the copro-ELISA’s sensitivity and specificity?

Please elaborate on when the questionnaire was administered. The paper states that the questionnaire was completed by the dog’s owner at the beginning of the study. However, some of the questions appear to be addressing collar compliance issues, which would not be available immediately. Was a questionnaire also administered to the control group? 

Please elaborate on the statistical methods used, including what specific variables were assessed. The statement, “…(GEE) was employed to analyze the data which were the categorical outcome variables obtained by continuous repeated measures” is not clear.

Much of the information located in the footnotes of tables 2-4 seems important enough to be included in the Methods text.

Reviewer #2: the work is clear, the objectives are well planned and the development is well articulated

**Results**

-Does the analysis presented match the analysis plan?

-Are the results clearly and completely presented?

-Are the figures (Tables, Images) of sufficient quality for clarity?

Reviewer #1: What is meant by the “dogs’ incompatibility with the collar’?

What happened to dogs whose owners elected to remove the collar during the study period (e.g., if an owner removed the collar at 6 months, were they given instructions to manually deworm the dog for the remainder of the study)? 

Table 1- Please explain the missing numbers column. Were there any dogs lost to follow-up during the study?

Table 4- How can the authors be sure that the dogs actually ingested the praziquantel in 10 minutes or 30 minutes? Also, what is meant be “dogs’ reaction rate”? What is a deworming reminder?

Can you provide additional information about the dogs that became infected during the study? Did you find new dogs infected at each sampling or were there dogs that were positive over multiple time points? 

What was done with the positioning data?

Reviewer #2: the results are clearly and completely presented

**Conclusions**

-Are the conclusions supported by the data presented?

-Are the limitations of analysis clearly described?

-Do the authors discuss how these data can be helpful to advance our understanding of the topic under study?

-Is public health relevance addressed?

Reviewer #1: How often do the collars need to be recharged? 

The designation of part of this study as a “prospective cohort field study with random sampling” is a bit unclear. 

Please discuss study limitations as well as collar limitations. How expensive are the collars? How do you plan to address non-compliance issues? Do you see the bulk and weight of the collar being problematic?

Reviewer #2: the conclusions are vsupported by the data

**Editorial and Data Presentation Modifications?**

Reviewer #1: Based on the abstract, it is not clear what the smart collar actually does (i.e., the authors need to define “smart-collar deworming”).

I don’t understand the use of the term “cross-species”. Perhaps just use the term zoonotic. 

The authors indicate that the global burden of CE is approximately 1 million. One million what? 

Dog deworming is likely to have little impact on the E. multilocularis wildlife cycle, which should be acknowledged.

If future versions of this paper are submitted, inclusion of page numbers would be helpful.

Reviewer #2: minor revisions

**Summary and General Comments**

Reviewer #1: This is largely a proof-of-concept study looking at a dog collar that dispenses praziquantel baits. After reading the paper, I’m still not entirely sure how the collar works. Based on figure 1 (and the provided video), it appears that the collar is a smart pill dispenser that drops a praziquantel bait at predetermined intervals. However, the authors need to include additional information in the text about how the collar functions.

Reviewer #2: The limitations of the use of the PZQ have been well developed in Larrieu and Zanini

It would be interesting to include some description of the substrate of the baits for the dogs to eat, given the bad smell and taste of PZQ

Although the manuscript is very detailed, it would be good to have some data on the variability of the weight of the dogs and the consequent estimation of the doses (1 or 2 pills) and consequently the estimation of the number of possible deworming.

A clearer description of the RMS and how deworming is activated would be helpful

PLOS authors have the option to publish the peer review history of their article (what does this mean?). If published, this will include your full peer review and any attached files.

Reviewer #1: No

Reviewer #2: Yes: Larrieu Edmundo
---

## [Decision Letter · Decision Letter 1]

6 Apr 2021

Dear Zhou,

Thank you very much for submitting your manuscript "Smart Deworming Collar: A Novel Tool for Reducing Echinococcus Infection in Dogs" for consideration at PLOS Neglected Tropical Diseases. As with all papers reviewed by the journal, your manuscript was reviewed by members of the editorial board and by several independent reviewers. The reviewers appreciated the attention to an important topic. Based on the reviews, we are likely to accept this manuscript for publication, providing that you modify the manuscript according to the review recommendations. 

Please ensure the remaining issues identified by one of the reviewers are addressed

Sincerely,

Paul R. Torgerson

Associate Editor

Adriano Casulli

Deputy Editor

Please ensure the remaining issues identified by one of the reviewers are addressed

Reviewer's Responses to Questions

**Key Review Criteria Required for Acceptance?**

**Methods**

-Are the objectives of the study clearly articulated with a clear testable hypothesis stated?

-Is the study design appropriate to address the stated objectives?

-Is the population clearly described and appropriate for the hypothesis being tested?

-Is the sample size sufficient to ensure adequate power to address the hypothesis being tested?

-Were correct statistical analysis used to support conclusions?

-Are there concerns about ethical or regulatory requirements being met?

Reviewer #1: (No Response)

Reviewer #2: The paper is now clear and precise.

**Results**

-Does the analysis presented match the analysis plan?

-Are the results clearly and completely presented?

-Are the figures (Tables, Images) of sufficient quality for clarity?

Reviewer #1: (No Response)

Reviewer #2: perfect

**Conclusions**

-Are the conclusions supported by the data presented?

-Are the limitations of analysis clearly described?

-Do the authors discuss how these data can be helpful to advance our understanding of the topic under study?

-Is public health relevance addressed?

Reviewer #1: (No Response)

Reviewer #2: clear

**Editorial and Data Presentation Modifications?**

Reviewer #1: (No Response)

Reviewer #2: (No Response)

**Summary and General Comments**

Reviewer #1: Thank you for clarifying most of my questions. My remaining comments are below. 

Please use care when using of the term “rate”. In most instances, I believe the authors mean “proportion”.

Please provide the actual sensitivity and specificity of the copro-ELISA rather than >90%. 

It is still not clear if the information obtained from the manual deworming group (e.g., questions regarding whether the dog was dewormed and proof of deworming) was collected using a standardized questionnaire.

I’m still a bit confused about how bait consumption was confirmed. I understand how this was conducted by the researchers upon initial attachment of the collar. However, could the authors please clarify how dog owners recorded whether or not a dog in the smart collar group ingested the praziquantel bait within 10 or 30 minutes? Were owners asked to observe the dogs at 6:00 a.m. on mornings when the bait was dispensed? Did the owners record this information in writing? Without a defined protocol, I’m a bit skeptical of these values. 

The text indicates that 741 dogs were sampled at baseline (line 269). However, table 1 indicates that only 687 dogs were sampled at time “0”. Since dogs were “missing” at all sampling time points, please clarify when the collars were placed relative to sample collection.

Additional editing for language would be helpful. However, I will leave this to the Editor’s discretion.

Reviewer #2: (No Response)

PLOS authors have the option to publish the peer review history of their article (what does this mean?). If published, this will include your full peer review and any attached files.

Reviewer #1: No

Reviewer #2: Yes: Edmundo Larrieu

Figure Files:

Data Requirements:

Reproducibility:

References

---

## [Editor Report · Decision Letter 2]

4 May 2021

Dear Zhou,

We are pleased to inform you that your manuscript 'Smart Deworming Collar: A Novel Tool for Reducing Echinococcus Infection in Dogs' has been provisionally accepted for publication in PLOS Neglected Tropical Diseases.

Best regards,

Paul R. Torgerson

Associate Editor

Adriano Casulli

Deputy Editor

---

## [Editor Report · Acceptance letter]

15 Jun 2021

Dear Prof. Zhou,

We are delighted to inform you that your manuscript, "Smart Deworming Collar: A Novel Tool for Reducing Echinococcus Infection in Dogs," has been formally accepted for publication in PLOS Neglected Tropical Diseases.

Best regards,

Shaden Kamhawi

co-Editor-in-Chief

Paul Brindley

co-Editor-in-Chief
